# Comparison of the Safety and Immunogenicity of FAKHRAVAC and BBIBP-CorV Vaccines when Administrated as Booster Dose: A Parallel Two Arms, Randomized, Double Blind Clinical Trial

**DOI:** 10.3390/vaccines10111800

**Published:** 2022-10-26

**Authors:** Mohammadreza Ahi, Ramin Hamidi Farahani, Pouria Basiri, Ahmad Karimi Rahjerdi, Ali Sheidaei, Kimiya Gohari, Zahra Rahimi, Fatemeh Gholami, Milad Moradi, Farzad Ghafoori Naeeni, Kosar Naderi Saffar, Soheil Ghasemi, Babak Barati, Sohrab Moradi, Arina Monazah, Fatemeh Pouranvari, Mohsen Forooghizadeh

**Affiliations:** 1Clinical Trial Center of Iran University of Medical Sciences (IUMS), Tehran P.O. Box 14535, Iran; 2AJA University of Medical Sciences, Tehran P.O. Box 1411718541, Iran; 3Stem Cell Technology Research Center (STRC), Tehran P.O. Box 1997775555, Iran; 4Milad Daro Noor Pharmaceutical (MDNP) Company, Tehran P.O. Box 1986936914, Iran; 5Department of Passive Defence, Malek Ashtar University of Technology, P.O. Box Tehran 1678815611, Iran

**Keywords:** immunogenicity, safety, FAKHRAVAC^®^, BBIBP-CorV, clinical trial, neutralizing antibody (Nab) and booster-dose injection

## Abstract

*Purpose*: This study was completed to assess the immunogenicity and safety of the FAKHRAVAC and BBIBP-CorV vaccines as a booster dose in the population with a history of receiving two doses of BBIBP-CorV vaccine. *Methods*: In this double-blind, parallel clinical trial, we randomly assigned healthy adults with a history of receiving two doses of the BBIBP-CorV vaccine, who then received either the FAKHRAVAC or BBIBP-CorV vaccine as a booster dose. The trial is registered in the Iranian Registry of Clinical Trial document depository (Code: IRCT20210206050259N4). *Results*: The outcomes that were monitored in this study were serum neutralizing antibody (Nab) activity, immunoglobulin G (IgG) level, local and systemic adverse reactions, serious adverse events, suspected unexpected serious adverse reactions, and medically attended adverse events. After administering vaccines to 435 participants, the most frequent local and systemic adverse reactions were tenderness and nausea in 23.7% and 1.4% of cases, respectively. All adverse events were mild, occurred at a similar incidence in the two groups, and were resolved within a few days. *Conclusions*: On the 14th day after the booster dose injection, the seroconversion rate (i.e., four-fold increase) of Nabs for seronegative participants were 87% and 84.6% in the FAKHRAVAC^®^ and BBIBP-CorV groups, respectively. This study shows that the FAKHRAVAC^®^ vaccine, as a booster dose, has a similar function to the BBIBP-CorV vaccine in terms of increasing the titer of virus-neutralizing antibodies, the amount of specific antibodies, and safety.

## 1. Introduction

After the outbreak of the SARS-CoV-2 Delta strain (B.1.617.2) and decreased protection against COVID-19 infection, scientists have started to assess the implications of booster dose vaccination against COVID-19 in reducing both viral transmission and disease fatality [1]. There is a disagreement among researchers as to whether the homologous or heterologous method is used to select the booster dose of the COVID-19 vaccine. A trial in Sweden shows that a heterologous boost injection has a more immunogenic effect than a homologous schedule [2]. In addition, higher T-cell responses have been demonstrated in heterologous plans, compared with homologous ones [3]. On the other hand, a study about the m1273 vaccine suggests using the homologous method with a half dose (50 μg), when used as a booster [4]. In any case, the immunogenicity of the homologous or heterologous booster dose was superior to the control group, regardless of which vaccine had been received in the primary vaccination [5].

The BBIBP-CorV (also known as Sinopharm or BIBP, developed by Sinopharm’s Beijing Institute of Biological Products: BIBP) is one of the first vaccines listed in EUL [6], which has been approved in 91 countries and also tested in 30 trials among 12 countries [7]. The FAKHRAVAC^®^ is an inactivated vaccine with seeds isolated from oropharynx swabs of confirmed COVID-19 patients admitted to Iranian hospitals, which was developed by Milad Darou Noor Pharmaceutical Company (MDNP) [8]. This vaccine is currently licensed for emergency use in Iran, and its validation process via a third-phase clinical trial is developing to the later steps. This study aimed to answer whether a vaccine with the same platform can be used as a booster dose. In the following section, we will investigate the reactogenicity and immunogenicity of the FAKHRAVAC and BBIBP-CorV vaccines as a booster dose after the administration of two doses of BBIBP-CorV.

## 2. Methods

### 2.1. Study Design

We conducted a randomized, double-blind clinical trial with two parallel groups of healthy adults to compare the immunogenicity and safety of the FAKHRAVAC and BBIBP-CorV vaccines as a booster dose in the population with a history of receiving two doses of the BBIBP-CorV vaccine.

Based on the assigned random code, a booster dose (either FAKHRAVAC^®^ or BBIBP-CorV) was injected into the deltoid muscle of the volunteers. The participants were monitored for half an hour and their vital signs were checked. Furthermore, the clinical history of all the participants was filed via telephone monitoring for six months to see if there were any adverse events. Sampling was performed on the 14th day, after receiving a booster dose from all volunteers in both groups, to assess the immunogenicity of the booster dose.

The study protocol, available in Appendix A, was approved by the National Research Ethics Committee (approval number IR.NREC.1400.014, the 28th of November 2021) and registered in the Iranian Registry of Clinical Trial document depository (Code: IRCT20210206050259N4).

### 2.2. Participants

Participants were invited to the clinical trial study through a website. Volunteers who accepted the terms and conditions of the informed consent form (Appendix A) were asked to complete the initial screening questionnaire online. Candidates who were found eligible initially were invited to the trial center to sign a hard-copy document of informed consent and then were further evaluated to see how they met the inclusion criteria. The inclusion criteria that must be met were: being the age of 18 or older, not having a history of COVID-19 (since receiving the last dose of primary vaccination with BBIBP-CorV), being fully vaccinated and attended the trial within 75 to 195 days after vaccination, not being pregnant, and obtaining the signed informed consent form. The participants that met the exclusion criteria were the ones who had a history of allergy to drugs or vaccines, current acute or chronic symptomatic illness that required ongoing medical or surgical care, history of severe cardiovascular disease, lactation, history of receiving any vaccine during the 14 days period prior to the day of receiving the booster dose, history of transfusion of any blood product or immunoglobulin within the three months before receiving booster dose, history of diseases that resulted in immunosuppression (suspected or definite), history of long-term use of immunosuppressive drugs or systemic corticosteroids in the last four months leading up to the screening day, history of diagnosis or treatment for cancer (except basal cell carcinoma and cervical carcinoma in-situ), history of uncontrolled psychiatric severe illnesses, history of blood disorders, history of chronic neurological diseases, current drug/alcohol abuse (addiction), acute febrile illness at the time of booster vaccine injection, having splenectomy (for any reason), having any close contact with a definitively infected person with COVID-19 within the two weeks before the day of receiving the booster dose, and current use of anticoagulants and antiplatelet agents.

### 2.3. Randomization and Masking

This study utilized the variable size permuted block randomization method. The participants were randomly allocated into two groups who received the FAKHRAVAC^®^ or BBIBP-CorV booster dose. The randomization codes were generated by the rand function in the Microsoft Excel^®^ 2016 software. A non-repeating five-character (randomization code) was assigned to each participant.

Implementation of blinding was completed by the person responsible for the injection. Once the participant became eligible to receive the vaccine, a randomization code was assigned to the volunteer. The vaccine type was displayed on the screen of the vaccinator until the vaccination was confirmed. Thus, all participants, investigators (except the person in charge of preparing and inoculating the vaccine), laboratory staff, and statisticians were blinded to group allocations.

### 2.4. Procedures

After assessing vital signs and conducting a physical examination on day zero, each participant received the FAKHRAVAC or BBIBP-CorV vaccine based on the randomization sequence. Participants’ vital signs, local (pain, tenderness, swelling, and redness) and systemic (nausea/vomiting, diarrhea, headache, fatigue, and myalgia) adverse reactions were checked and filed daily for the first seven days and weekly up to one month after injection, via having a phone conversation with the participant. All participants were invited to visit the research center 14 and 90 days after receiving a booster dose to assess vaccines’ immunogenicity by collecting blood samples.

### 2.5. Outcomes

The primary outcome was assessing serum neutralizing antibody (Nab) activity in the second week and the third month after booster dose injection. The Nab of the participants’ serum, which was titered against SARS-CoV-2, was measured using a live virus microneutralization assay.

The secondary outcomes were safety profile (including reactogenicity, local and systemic adverse reactions), specific IgG antibody levels, medically attended adverse events (MAAE), serious adverse events (SAE), and suspected unexpected serious adverse reactions (SUSAR). Participants were asked to report their daily conditions by completing a web application questionnaire seven days after injection. During this period, the research team made daily telephone contacts with study participants who did not fill out the web application, and all the items in the questionnaire were checked. Study participants could contact resident physicians at the center 24/7 in case of any need for medical assistance or consult. Further, any possible medical conditions they might have experienced until the end of the study were filed. The IgG levels of the participants’ serum for SARS-CoV-2 N and S1RBD antigens were measured by an enzyme-linked immunosorbent assay (ELISA) method.

### 2.6. Statistical Analysis

The estimated sample size in this study was 200 persons per group (a total of 400 persons), and the baseline comparison was performed to examine the homogeneity between the study groups. In order to calculate the sample size in this study, Equation (1) was applied [9].
(1)$$N=\frac{[p_{a}\left(1-p_{a}\right)+p_{b}\left(1-p_{b}\right)]\;{\left(z_{1-\beta }+z_{1-\alpha }\right)}^{2}}{\left[(p_{a}-p_{b}\right)-d{]}^{2}}$$

According to the primary outcome of this study, the amount of *p_a_* and *p_b_* in Equation (1) was set as the percentage of participants with positive seroconversion on day 14 and 0 in the Sinopharm and FAKHRAVAC^®^ groups, respectively. Considering the time of conducting this study, the percentage of this vaccine’s efficacy (the closest available data) was used. In the Sinopharm group, the efficiency was considered to be around 80% [10,11]. In the third phase of the FAKHRAVAC^®^ trial, the efficiency of this vaccine was estimated to be 81.6%. Using the above formula and considering *α* = 2.5%, *β* = 80%, and *d* = 10%, the sample size was obtained as what is shown in Equation (2).
(2)$$N_{A}=\frac{\left[0.80\left(1-0.80\right)+0.816\left(1-0.816\right)\right]\;{\left(0.84+1.96\right)}^{2}}{\left[(0.80-0.816\right)-0.1{]}^{2}}=181$$

The total sample size obtained was 181 people in each group. Considering the fact that some of the volunteers would be excluded during trial, more than 181 persons were recruited for each group of study. Finally, a total of 435 people (216 people for the FAKHRAVAC^®^ group and 219 people for the Sinopharm group) were included in this study.

We monitored the indices of safety and immunogenicity in all participants who received booster doses and visited the research center for blood sampling the second week after injection. For the immunogenicity assessment, we calculated the geometric mean titers (GMT), geometric mean ratio (GMR), geometric mean fold increase (GMFI), and corresponding 95% Cis, which was based on a standard normal distribution of the log-transformation antibody titer and the variable “seroconversion rate”. The data were then analyzed by Stata 14.2 (Stata Corporation, College Station, TX, USA).

## 3. Results

From the 1st of December 2021 to the 17th of January 2022, 435 eligible participants were recruited for the trial. The participants were randomly assigned to the FAKHRAVAC^®^ group (*n* = 216) or the BBIBP-CorV group (*n* = 219). The mean ages of the FAKHRAVAC^®^ and the BBIBP-CorV groups were 41 and 42, respectively. In this study, 85.1% (370/435) of participants were male, and 14.9% were female, and these rates were the same in the two groups. The participants’ baseline characteristics and comorbidity were broadly similar across the groups (Table 1). Further, the results showed an almost uniform distribution of participants regarding the level of basal-specific antibodies in the random arms. The geometric mean of the specific antibody levels in the FAKHRAVAC^®^ group was slightly higher, which was not a significant difference (Table 2).

The activity of the Nab in the baseline was undetectable (titer < 4) in 25.98% of participants (113 out of 435). They did not have levels of protective antibodies against the COVID-19 despite receiving two doses of the BBIBP-CorV vaccine (on average, four months before the baseline). Fourteen days after administration of the booster dose, the seroconversion rate of Nab in seronegative participants was 87% and 84.6% in FAKHRAVAC^®^ and BBIBP-CorV groups, respectively. The GMFI of the seronegative participants was 24.1 (95% CI: 15.2–38.1) and 26.2 (95% CI: 16.4–42) two weeks after administering the booster dose in the FAKHRAVAC and BBIBP-CorV groups, respectively (Table 3).

The serum neutralizing activity in all participants had a GMT of 84.4 (95% CI: 70.5–101.1) and 72.6 (95% CI: 59.3–88.9) in the FAKHRAVAC^®^ and BBIBP-CorV groups two weeks after booster injection, respectively. The geometric means ratio of the FAKHRAVAC^®^ group compared to the BBIBP-CorV group on day 14 was 1.24. Fourteen days after the booster dose administration, the Nab seroconversion rate was 40.1% and 39.8% in the FAKHRAVAC^®^ and BBIBP-CorV groups, respectively (Table 4).

The geometric mean of antibodies against the S1RBD antigen was significantly boosted in both groups. The GMFI index showed this increment was 1.6 (1.41–1.79) and 2.1 (1.81–2.39) in the FAKHRAVAC^®^ and BBIBP-CorV groups, respectively. In addition, the S1RBD antibody index value changed from negative to positive (according to the manufacturers’ cut-off) in 79.31% and 82.93% of the samples, acquired from participants of the FAKHRAVAC^®^ and BBIBP-CorV groups, respectively (Table 5).

The safety study after booster dose administration showed neither SUSAR nor SAE. The most common local adverse event was tenderness (in 28.8% of BBIBP-CorV participants and 18.5% of FAKHRAVAC^®^ participants). In some cases, participants reported pain, swelling/stiffness, and redness (Figure 1A). In addition, the most commonly reported systemic reaction was nausea (in 0.9% of BBIBP-CorV participants and 1.9% of FAKHRAVAC^®^ participants). Participants also reported headaches, fatigue, and none of the participants reported diarrhea and myalgia (Figure 1B).

## 4. Discussion

This study shows that the FAKHRAVAC^®^, as a booster dose, has a similar function to the BBIBP-CorV vaccine in terms of increasing the titer of virus-neutralizing antibodies, the amount of specific antibodies, and safety. Therefore, the FAKHRAVAC^®^ can be used as a booster dose injection alternative to the BBIBP-CorV vaccine.

Because of emerging variants of concern (VOCs), the evaluation of various allocation strategies of COVID-19 vaccines and antiviral drugs has been tailored [12,13,14]. Meanwhile, heterologous vaccination with any available vaccine as the additional dose is confirmed as one of the promising strategies against emerging VOCs, such as Omicron [14]. Due to the short interval before the start of booster dose programs of vaccination against COVID-19 disease, little information is available about the safety and effectiveness of booster doses of different vaccines. However, all available information confirms the positive effect of the booster dose injection [5,15,16].

The reactions to vaccine administration, such as the emergence of abnormal vital signs, anaphylaxis, AEs, SAEs, SUSARs, MAAE, and local or systemic adverse events, were found to be insignificant in both groups of study. A total of 79 (36% of volunteer population) and 103 (48% of volunteer population) cases of local adverse events were observed in the Sinopharm and FAKHRAVAC^®^ groups, respectively. The occurrence of local adverse events of the Sinopharm vaccine in the current study (36%) is in agreement with previous studies (37.4%) [11], which is not significantly different from the prevalence of local adverse events in the FAKHRAVAC^®^ group (Appendix A). All of these events were mild (grade 1 and 2) and transient (i.e., disappearance of symptoms after one day), which confirm the safety of both vaccines. Furthermore, local events such as pain sting are more due to the intervention of the needle injection than the nature of the injected substance. Therefore, mild cases (such as grade 1 and 2 of pain) would not question the safety of the vaccine. Since the ethical permission of this study was issued to review the serological data only for 14 days after the injection of the booster dose, it was not possible for the researchers to further review the safety of the vaccine. However, safety data were collected actively for one month and by the self-declaration of volunteers for six months.

Baseline assessments in our study show that 25.98% of participants who had previously received two doses of the BBIBP-CorV vaccine did not have an adequate protection level of Nab. Nabs’ seroconversion rates (four-fold increase) were 87% and 84.6% in the seronegative populations of FAKHRAVAC^®^ and BBIBP-CorV, respectively. In addition, the value of the S1RBD antibody index changed from negative to positive in 79.3% and 82.9% of the FAKHRAVAC^®^ and BBIBP-CorV groups, respectively. Considering these results, a comparison of the FAKHRAVAC^®^ and BBIBP-CorV in seronegative participants supports the conclusion that both vaccines will increase immunity against COVID-19 to a protective level.

For all participants, the GMT of Nab boosted significantly, while the GMFI index increased by more than 6-fold and 7-fold in the FAKHRAVAC^®^ and BBIBP-CorV groups, respectively. Furthermore, Nabs’ seroconversion rates (four-fold increase) were 40.11% and 39.77% in the FAKHRAVAC^®^ and BBIBP-CorV groups, respectively. In addition, the GMFI index value shows that the GM of specific antibodies against the S1RBD antigen increased by 1.59 (1.41–1.79) and 2.08 (1.81–2.39) times compared to the baseline in the FAKHRAVAC^®^ and BBIBP-CorV vaccine groups. Considering both the fact that the BBIBP-CorV group rated a higher GM of the S1RBD antibody level than the FAKHRAVAC^®^ group, and the other fact that both vaccines resulted in the same level of neutralizing antibody, one can say there are other pathways (than S1RBD), which are involved in immune protection, made by FAKHRAVAC^®^.

## Figures and Tables

**Figure 1 vaccines-10-01800-f001:**
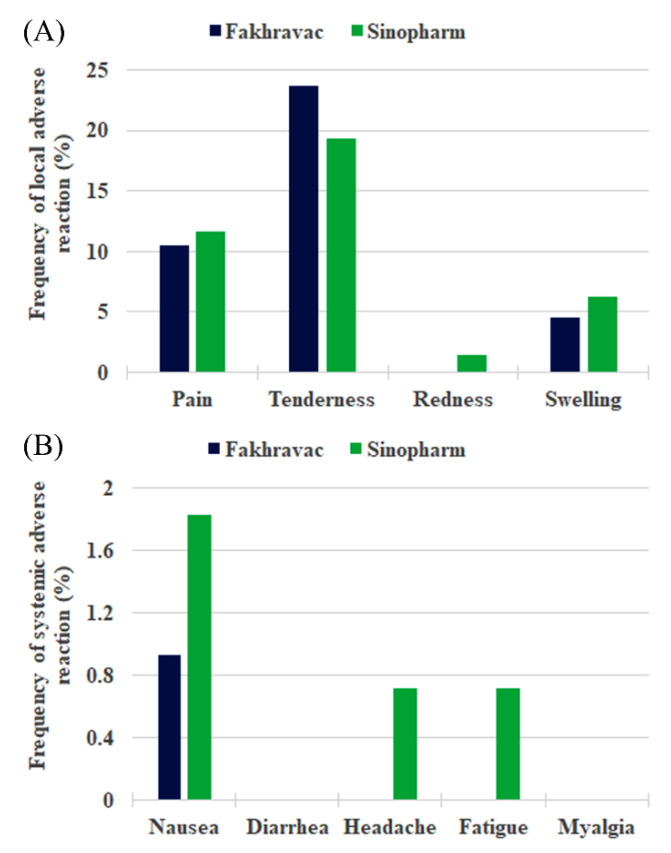
Percentage of participants with adverse reactions within 14 days of vaccines post-injection. (**A**,**B**) represent the percentage of “local” and “systemic” adverse reactions, respectively.

**Table 1 vaccines-10-01800-t001:** Baseline characteristics.

Items.		BBIBP-CorV	FAKHRAVAC	Total
		(*n* = 219)	(*n* = 216)	(*n* = 435)
Sex	Male	188(85.84%)	182(84.26%)	370(85.06%)
	Female	31(14.16%)	34(15.74%)	65(14.94%)
Median (IQR)		42(18)	41(16.5)	41(18)
	18–30	26(11.87%)	34(15.74%)	60(13.79%)
Age range	30–40	65(29.68%)	65(30.09%)	130(29.89%)
(Percentage of population: %)	40–50	59(26.94%)	63(29.17%)	122(28.05%)
	50–60	37(16.89%)	31(14.35%)	68(15.63%)
	>60	32(14.61%)	23(10.65%)	55(12.64%)
Body mass index; Mean (SD)		26.11(3.71)	26.56(3.90)	26.33(3.80)
Education; *n* (%)	No formal education	11(5.02%)	7(3.24%)	18(4.14%)
	Up to high school Diploma	15(6.85%)	26(12.04%)	41(9.43%)
	High school Diploma	55(25.11%)	48(22.22%)	103(23.68%)
	Diploma plus	33(15.07%)	29(13.43%)	62(14.25%)
	BSc	79(36.07%)	81(37.50%)	160(36.78%)
	MSc	18(8.22%)	12(5.56%)	30(6.90%)
	PhD and above	8(3.65%)	13(6.02%)	21(4.83%)

**Table 2 vaccines-10-01800-t002:** Baseline geometric mean of the level of specific antibodies.

Antibodies	BBIBP-CorV (*n* = 219)	FAKHRAVAC (*n* = 216)	Total (*n* = 435)
Antibody for S1RBD Ag, GM	2.46	3.02	2.73
Antibody for N Ag, GM	2.37	3.19	2.74
Neutralizing antibody activity, GMT	17.84	26.09	21.54

**Table 3 vaccines-10-01800-t003:** Serum neutralizing antibody indices in seronegative group. ^a^ The geometric mean of the amount of antibody in each vaccine was divided by the amount of antibody in the baseline (i.e., Ref) day in the same vaccine group. ^b^ The geometric mean of the amount of antibody in each vaccine was divided by the amount of antibody in the BBIBP-CorV (Ref) group on the same day. ^c^ The ratio of the geometric mean changes of the two vaccines from the baseline (Ref) value was divided by the ratio of the changes in the BBIBP-CorV (Ref) group of baseline values.

Indices		Baseline	Day 14
GMT (95% CI)	BBIBP-CorV (*n* = 52)	2.4 (2.0, 2.9)	37.8 (26.7, 53.4)
	FAKHRAVAC (*n* = 46)	2.6 (2.1, 3.3)	40.7 (29.8, 56.2)
GMFI (95% CI) ^a^	BBIBP-CorV (*n* = 52)	1.0 (Ref)	26.18 (16.31–42.04, *n* = 52)
	FAKHRAVAC (*n* = 46)	1.0 (Ref)	24.08 (15.23–38.06, *n* = 46)
GMR (95% CI) ^b^	BBIBP-CorV (*n* = 52)	1.0 (Ref)	1.0 (Ref)
	FAKHRAVAC (*n* = 46)	1.15 (0.95–1.38, *n* = 53)	1.08 (0.57–2.03, *n* = 46)
GMFR (95% CI) ^c^	BBIBP-CorV (*n* = 52)	1.0 (Ref)	1.0 (Ref)
	FAKHRAVAC (*n* = 46)	1.0 (Ref)	0.94 (0.52–1.7, *n* = 46)
Four-fold rise; *n* (%)	BBIBP-CorV (*n* = 52)	-	44 (84.62%)
	FAKHRAVAC (*n* = 46)	-	40 (86.96%)

**Table 4 vaccines-10-01800-t004:** Serum neutralizing antibody (Nab) indices in all participants. The GMT, GMFI, GMR, and GMFR are explained in the caption of Table 1. Superscripts of a, b, c represent the geometric mean of the amount of antibody in each vaccine, which was divided by the either amount of antibody in the baseline (Ref) day in the same vaccine group, the BBIBP-CorV (Ref) group on the same day, and changes of the two vaccines from the baseline (Ref), respectively.

Indices	Baseline	Day 14
BBIBP-CorV (*n*= 52)	17.84 (13.78–23.11, *n* = 218)	72.6 (59.27–88.94, *n* = 176)
FAKHRAVAC (*n* = 46)	26.09 (20.09–33.89, *n* = 214)	84.37 (70.45–101.05, *n* = 177)
BBIBP-CorV (*n* = 52)	1.0 (Ref)	7.03 (4.84–10.21, *n* = 176)
FAKHRAVAC (*n* = 46)	1.0 (Ref)	6.07 (4.44–8.28, *n* = 177)
BBIBP-CorV (*n* = 52)	1.0 (Ref)	1.0 (Ref)
FAKHRAVAC (*n* = 46)	1.73 (1.02–2.94, *n* = 214)	1.24 (0.84–1.83, *n* = 177)
BBIBP-CorV (*n* = 52)	1.0 (Ref)	1.0 (Ref)
FAKHRAVAC (*n* = 46)	1.0 (Ref)	0.79 (0.49–1.28, *n* = 177)
BBIBP-CorV (*n* = 52)	-	70 (39.77%)
FAKHRAVAC (*n* = 46)	-	71 (40.11%)

**Table 5 vaccines-10-01800-t005:** Specific IgG antibody against S1RBD antigen indices in all participants. ^a^ The geometric mean of the amount of antibody in each vaccine was divided by the amount of antibody in the baseline (Ref) day in the same vaccine group. ^b^ The geometric mean of the amount of antibody in each vaccine was divided by the amount of antibody in the BBIBP-CorV (Ref) group on the same day. ^c^ The ratio of the geometric mean changes of the two vaccines from the baseline (Ref) value was divided to the ratio of the changes in the BBIBP-CorV (Ref) group of baseline values.

Items	BBIBP-CorV	FAKHRAVAC
Baseline GM (95% CI)	2.46 (2.14–2.83, *n* = 219)	0.02 (2.65–3.45, *n* = 216)
GM (95% CI, *n*) at day 14	5.23 (4.83–5.67, *n* = 179)	4.67 (4.23–5.14, *n* = 181)
GMFI (95% CI) at day 14 ^a^	2.08 (1.81–2.39, *n* = 179)	1.59 (1.41–1.79, *n* = 181)
GMR (95% CI) at day 14 ^b^	1 (Ref)	0.89 (0.79–1.01, *n* = 181)
GMFR (95% CI) at day 14 ^c^	1 (Ref)	0.74 (0.62–0.89, *n* = 181)
Seroconversion, *n*/N (%)	34/41 (82.93%)	23/29 (79.31%)

## Data Availability

The study did not report any data.

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
