# Peer review of "Comparison of the Safety and Immunogenicity of FAKHRAVAC and BBIBP-CorV Vaccines when Administrated as Booster Dose: A Parallel Two Arms, Randomized, Double Blind Clinical Trial"

_vaccines, 2022, doi:10.3390/vaccines10111800_

Round 1

Reviewer 1 Report

This study compared the safety and immunogenicity of FAKHRAVAC and BBIBP-CorV vaccines when administrated booster doses. The study design is appropriate, and the finding is convincing. It can be published.

Author Response

Response to Reviewer 1 Comments 

Point 1: This study compared the safety and immunogenicity of FAKHRAVAC and BBIBP-CorV vaccines
when administrated booster doses. The study design is appropriate, and the finding is convincing. It can be
published. English language and style are fine/minor spell check required 

Response 1: Minor spell check is done, which is highlighted using track change mode. 

Reviewer 2 Report

Ahi et al. performed a comparative study on the safety and immunogenicity of FAKHRAVAC and BBIBP-CorV vaccines when administered as a booster dose for population which has received two doses of BBIBP-CorV vaccine. The authors noted that both vaccines, which were used as a booster, had similar functions in increasing the titer of virus-neutralizing antibodies, the amount of specific antibodies, and safety. This is an important study and I would recommend its acceptance for publication. 

Author Response

Response to Reviewer 2 Comments 
Point 2: This study compared the safety and immunogenicity of FAKHRAVAC and BBIBP-CorV vaccines
when administrated booster doses. The study design is appropriate, and the finding is convincing. It can be
published. English language and style are fine/minor spell check required 

Response 1: Minor spell check is done, which is highlighted using track change mode. 

Reviewer 3 Report

Review comments

The research article by titled Mohammadreza Ahi et al “Comparison of the safety and immunogenicity of 2 FAKHRAVAC and BBIBP-CorV vaccines when administrated 3 as booster dose: A parallel two arms, randomized, double blind 4 clinical trial” is an interesting study, However, some of my major concerns are:

1.     The clinical trail participants were too low to be approved. I Wish the authors can provide more data by now as it has been around 10 months since January 2022.

2.     The data in Figure 1 shows that after a booster dose more than 10 percent of pain is a huge number. Also, other side effects such as tenderness are very high which include swelling and stiffness. The authors should provide more data from participants to confirm the safety of the FAKHRAVAC vaccine. 

Author Response

Response to Reviewer 3 Comments
Point 1: The research article by titled Mohammadreza Ahi et al “Comparison of the safety and immunogenicity of 2 FAKHRAVAC and BBIBP-CorV vaccines when administrated 3 as booster dose: A parallel two arms, randomized, double blind 4 clinical trial” is an interesting study, However, some of my major concerns are:
1. The clinical trail participants were too low to be approved. I Wish the authors can provide more data by now as it has been around 10 months since January 2022.
Response 1:
Regarding the first part of the point, the following formula and further explanations are added to section 2.9. (Statistical analysis) as follows:
“In order to calculate the sample size in this study, Equation 1 was applied [9].
Equation 1: N=[??(1−??)+??(1−??)] (?1−?+?1−∝)2[(??−??)−?]2
According to the primary outcome of this study, the amount of pa and pb in Equation 1 was set as the percentage of participants with positive seroconversion on day 14 and 0 in the Sinopharm and Fakhra groups, respectively. Considering the time of conducting this study, percentage of this vaccines efficacy (the closest available data) were used. In the Sinopharm group, the efficiency is considered to be around 80% [10]. In the third phase of Fakhra vaccine trial, the efficiency of this vaccine was estimated to be 81.6%. Using the above formula and considering ∝=2.5%, β=80% and d=10%, the sample size was obtained as what is shown in Equation 2.
Equation 2: ??=[0.80(1−0.80)+0.816(1−0.816)] (0.84+1.96)^2[(0.80−0.816)−0.1]2=181
The total sample size obtained is 181 people in each group. Considering this fact that some of the volunteers would be excluded during trial, more than 181 persons were recruited for each group of study. Finally, a total of 435 people (216 people for Fakhra group and 219 people for Sinopharm group) were included in this study.”
Besides, regarding the second part of the point, we have to pay attention that the ethical permission of this study was issued to review the serological data only for 14 days after the injection of the booster dose, and it was not possible for the researchers to further review; since you mentioned “provide more data by now as it has been around 10 months since January 2022”. However, safety data was collected actively for one month and by self-declaration of volunteers for six months. Therefore, the following sentences are added to the end of third paragraph of discussion as follows:
2
“Since the ethical permission of this study was issued to review the serological data only for 14 days after the injection of the booster dose, it was not possible for the researchers to further review the safety of the vaccine. However, safety data was collected actively for one month and by self-declaration of volunteers for six months.”
Point 2: The data in Figure 1 shows that after a booster dose more than 10 percent of pain is a huge number. Also, other side effects such as tenderness are very high which include swelling and stiffness. The authors should provide more data from participants to confirm the safety of the FAKHRAVAC vaccine.
Response 2: The following information is described in supplementary file 4:
According to Babaee et al.* study on adverse effects following COVID-19 vaccination in Iran, of 2653 participants who received two doses of COVID-19 vaccines, 55.56% reported adverse effects after the first dose and 35.7% after the second dose. Sputnik V caused the most adverse effects in 82.7% vaccine recipients, compared with 70.5% for AstraZeneca and 37.4% for the Sinopharm vaccine. Further, occurrence of adverse effects was significantly higher in participants with a history of COVID-19 infection who received the Sputnik V or AstraZeneca vaccine than those who either reported no history of infection or received Sinopharm vaccine (P = 0.001).
Comparison of Local adverse events in the current study (Ahi et al) with Babaee et al study.
*Babaee, E., Amirkafi, A., Tehrani-Banihashemi, A., SoleimanvandiAzar, N., Eshrati, B., Rampisheh, Z., Asadi-Aliabadi, M. and Nojomi, M., 2022. Adverse effects following COVID-19 vaccination in Iran. BMC Infectious Diseases, 22(1), pp.1-8.
These sentences are added to the third paragraph of discussion for more clarification:
A total of 79 (36% of volunteer population) and 103 (48% of volunteer population) cases of local adverse events were observed in the Sinopharm and Fakhravac groups, respectively. The occurance of local adverse events of Sinopharm vaccine in the current study (36%) is in agreement with previous studies (37.4%) [11], which is not significantly different from prevalence of local adverse events in Fakhravac group. All these events were mild (grade 1 and 2) and transient (i.e. disappearance of symptoms after one day), which confirm the safety of both vaccines. Besides, local events such as pain sting are more due to the intervention of the needle injection than the nature of the injected substance. Therefore, mild cases (such as grade 1 and 2 of pain) would not question the safety of the vaccine. Since the ethical permission of this study was issued to review the serological data
Sinopharm
(Babaee et al.)*
n= 1564
Fakhra
(Curent study)
n=216
Sinopharm
(Curent study)
n=219
585 (37.4%)
103 (48%)
79 (36%)
Local adverse reactions Pain
27 (12.5 %)
23 (11%)
Grade 1
1 (0.5 %)
0 (0%)
Grade 2 Tenderness
63 (29.5%)
40 (17%)
Grade 1 Redness
0 (0%)
3 (2%)
Grade 1 Swelling
12 (5.5%)
13 (6%)
Grade 1
3
only for 14 days after the injection of the booster dose, it was not possible for the researchers to further review the safety of the vaccine. However, safety data was collected actively for one month and by self-declaration of volunteers for six months.
Response 3: Recent references are added to the method section to provide sufficient background and make adequate description. Further sentences are added to more clarification of results.

Round 2

Reviewer 3 Report

The authors have answered my queries. It is suitable for acceptance after the final evaluation from editors. Thank you